

# Predictive performance of clinical scores and survival outcomes in critically ill patients with sepsis: a prospective longitudinal study at a tertiary medical centre in Ethiopia

Girum Tesfaye Kiya[1], Zeleke Mekonnen[1], Elsah Tegene Asefa[2], Edosa Kejela[3], Edosa Tadasa[1], Esayas Kebede Gudina[2], Tilahun Yemane[1] and Gemeda Abebe[1]

[1] School of Medical Laboratory Sciences, Jimma University, Jimma, Oromia, Ethiopia
[2] Department of Internal Medicine, Jimma University, Jimma, Oromia, Ethiopia
[3] Department of Anesthesia, Jimma University, Jimma, Oromia, Ethiopia

## ABSTRACT

**Background**. Clinical scores such as quick sequential organ failure assessment (qSOFA), early warning score (EWS), and universal vital assessment (UVA) are commonly used as screening and prognostic tools in sepsis. However, consistent findings across different regions and hospital settings remain limited. This study aims to evaluate the performance of these clinical scores and identify predictors of survival in septic adults admitted to the ICU.

**Methods**. A prospective longitudinal study was conducted from October 10, 2023, to October 9, 2024, involving adults admitted to the intensive care unit (ICU). Clinical scores were calculated on the first, third, and fifth days of admission. Their performance in detecting sepsis, identifying culture-positive cases, and predicting mortality was assessed using receiver operating characteristic (ROC) curves. Kaplan–Meier survival analysis was used to estimate survival probabilities, and a Cox proportional hazards model was applied to identify predictors of survival in sepsis patients.

**Results**. Of the ICU-admitted patients, 148 (51.9%) were septic, and 54 (36.5%) of them died in the ICU. The modified early warning score (MEWS) showed good performance in identifying sepsis (area under the curve (AUC) = 0.67; 95% confidence interval (CI) [0.61–0.73]) and culture-positive cases (AUC = 0.65; 95% CI [0.50–0.80]) on the day of admission. MEWS also performed better in predicting mortality on day five (AUC = 0.75; 95% CI [0.57–0.93]). Patients with sepsis had significantly lower survival probabilities than those without sepsis (log-rank test, $p < 0.001$). Each additional point in the Glasgow Coma Scale (GCS) score reduced the hazard of death by 10% (HR = 0.90; 95% CI [0.84–0.97]; $p = 0.005$).

**Conclusion**. The MEWS score outperformed other clinical scores in identifying sepsis, detecting culture-positive cases, and predicting mortality. Sepsis was associated with higher mortality, which decreased with increasing GCS scores. MEWS may support early identification of sepsis and mortality risk, and its use could help improve ICU survival through timely intervention.

Corresponding author
Girum Tesfaye Kiya,
tesfaye.girum@ju.edu.et,
girumtesfaye12@gmail.com

## INTRODUCTION

Sepsis is a dysregulated immune response to infection that causes life-threatening organ dysfunction (*Singer et al., 2016*). It contributes to one-third of intensive care unit (ICU) admissions globally and is responsible for 35% of ICU deaths. The mortality rate is even higher in Africa reaching 47% (*Sakr et al., 2018*). In sub-Saharan Africa, the prevalence of sepsis in ICUs has been reported at 31%, with a mortality rate of 46% (*Kiya et al., 2024*).

While early identification, and appropriate and timely management are the key steps to mitigate the burden of sepsis, achieving these goals remains a significant challenge (*World Health Organization, 2023*). This difficulty arises from the complex nature of sepsis pathogenesis, which involves factors related to pathogens and hosts. The condition often presents similarly across different pathogens and between infectious and non-infectious causes. To tackle these challenges, various clinical and laboratory tools have been developed to aid in diagnosis and mortality prediction of sepsis.

For over two decades, systemic inflammatory response syndrome (SIRS) served as the primary tool for screening and diagnosing sepsis (*Bone et al., 1992*). In 2016, sepsis was redefined as a life-threatening organ dysfunction caused by a dysregulated immune responses to infection (*Singer et al., 2016*). Organ dysfunction is expressed in terms of change in total sequential organ failure assessment (SOFA) score by 2 or more points. A SOFA score of 2 corresponds to an overall mortality risk of approximately 10% in a general hospital population with suspected infection. The SOFA score evaluates six key physiological system: respiratory, coagulation, liver, cardiovascular, central nervous system, and renal function. Additionally, the quick sequential organ failure assessment (qSOFA) score, which considers three clinical parameters—systolic blood pressure, mental status and respiratory rate—serves as a simplified screening tool for detecting organ dysfunction.

Moreover, the early warning scores (EWS), derived from patients' vital signs and mental status have been utilized to assess clinical deterioration of patients (*Chua, Rusli & Aitken, 2024*). Among these, the National Early Warning Score (NEWS) and the modified early warning score (MEWS) have been employed as screening tools and predictor of mortality in sepsis patients across various hospital wards and ICUs (*Qiu, Lei & Zhou, 2023*; *Khwannimit, Bhurayanontachai & Vattanavanit, 2019*; *Keep et al., 2016*; *Corfield et al., 2014*; *Liu et al., 2020*). For instance, the NEWS has demonstrated strong diagnostic and prognostic performance in sepsis patients at emergency units and ICUs (*Khwannimit, Bhurayanontachai & Vattanavanit, 2019*; *Corfield et al., 2014*; *Durr et al., 2022*; *Zhou, Lan & Bin, 2020*; *Hsieh et al., 2024*). A meta-analysis conducted in low- and middle-income countries found that MEWS and qSOFA had comparable predictive accuracy (*Adegbite et al., 2021*). Furthermore, this study highlighted the universal vital assessment (UVA) score—developed using multiple cohorts in sub-Saharan Africa—as having the highest sensitivity for predicting in-hospital mortality. Unlike other scoring systems, UVA incorporates

additional parameters such as oxygen saturation and human immunodeficiency virus (HIV) status (*Moore et al., 2017*).

Despite extensive research on the predictive performance of clinical scores for sepsis and mortality, findings remain inconsistent across various regions and hospital settings. In low-resource areas such as sub-Saharan Africa—where sepsis prevalence and related deaths are high—the effectiveness of these scores has not been thoroughly evaluated. Moreover, most previous studies have focused on general wards and emergency units, with limited data available on ICU mortality prediction. The progression of these scores over subsequent days of admission and their role in predicting mortality have also not been explored. Early risk stratification using accessible clinical and laboratory tools is crucial in ICU settings for making timely, informed decision in patient management. Additionally, there is a gap in research concerning survival probability and its predictors among ICU admitted sepsis patients in this region. Therefore, this study aims to evaluate the predictive performance of clinical scores and investigate survival probability and its determinants in septic adults admitted to the ICU.

## MATERIALS & METHODS

### Study design, period, and setting

A prospective longitudinal study was conducted from October 10, 2023 to October 9, 2024 at Jimma University Medical Center (JUMC). JUMC is one of the oldest hospitals in Ethiopia with 800 bed capacity and a catchment population of over 20 million people. JUMC has a total of 36 beds in the medical, surgical, and emergency ICUs, with an estimated annual admission of 300 patients.

### Population and eligibility criteria

All patients admitted to the ICUs of JUMC were the source population. The study population were patients admitted to medical, surgical, and emergency ICUs of JUMC during the study period and who met the inclusion criteria. All adult patients aged 18 years and older who were admitted to ICUs and stayed for 24 hrs were included. Patients or their guardians who were not volunteer to participate in the study were excluded.

### Data collection tools and study variables

A structured checklist and questionnaire were used to collect background information and clinical data of the patients. Demographic and clinical data including age, sex, admission diagnosis, site of infection, comorbidities, organ failure, interventions done, and days of before and in-ICU stay were collected. Vital signs, Glasgow coma scale, and oxygen saturation were collected at day 1, day 3, and day 5 of admission. For survival analysis, the primary outcome variable was time to death in ICUs.

### Sepsis diagnosis, vital sign measurement and clinical scores calculation

Sepsis diagnosis was made by an attending physicians adjudication using clinical, laboratory, and radiological data available within 24 hrs of admission. The surviving sepsis campaign

guideline recommends hospitals to select the most accurate and timely approach to sepsis screening that they can feasibly implement (*Surviving Sepsis Campaign, 2019*).

In the ICUs, temperature was measured primarily from the axilla using a mercury thermometer every 2 hrs, and the average of these readings was used as the daily temperature measurement. Biocare PM-900, EDAN iM70, and Bionics BPM-190 patient monitors were used to measure the heart rate, blood pressure, and oxygen saturation of the patients. Respiratory rate was measured from the mechanical ventilator if the patient was on mechanical ventilation or counting manually. All devices undergo baseline testing to identify any deviation in readings followed by adjustments using standardized reference signals when appropriate. Sensor verification, and functional checks were also undertaken to ensure the reliability of measurements. Either attending physicians or clinical nurses assessed the Glasgow Coma Scale, which involves evaluating eye responses, verbal responses, and motor responses, and then assigning scores for each measurement.

Clinical scores were computed from vital signs, mental status, and oxygen saturation of the patients collected at day 1, day 3, and day 5 of admission, based on their original definitions proposed for each score as presented in Table S1 (*Singer et al., 2016*; *Bone et al., 1992*; *Moore et al., 2017*; *Subbe et al., 2001*; *RCoP, 2012*). The clinical scores included were SIRS, qSOFA, UVA, MEWS, and NEWS. These scores are easily computed from routine clinical and laboratory findings in the ICU settings of resource limited areas.

## Statistical analysis

Descriptive statistics were employed, including medians and IQRs for non-normally distributed variables, as assessed by the Shapiro–Wilk test. There were no normally distributed continuous variables in the study. Categorical variables were presented as frequencies. To evaluate the differences between patients with sepsis and without sepsis, as well as survivors and non-survivors, the rank sum test was used for non-normally distributed continuous variables. The $\chi^2$ test of independence was used to compare proportions for binomial variables where all observed counts of cells in contingency table were 5 or greater; while the Fisher exact test was used for any counts fewer than 5. Kaplan–Meier survival analysis with Log-Rank test was employed to see the survival probability between patients with sepsis and without sepsis. Cox proportional hazard model was run to identify predictors of survival in sepsis patients. Missing data were handled accounting for censored cases by including them in calculation up to their last known time. Performance characteristics for predicting in-ICU death, and recognizing sepsis and blood culture positivity were calculated for each score at each day of admission. This included sensitivity, specificity, likelihood ratios, and the area under the curve (AUC). All statistical analyses were performed using Stata, version 17.

## Ethics approval and consent to participate

This study was approved by the Jimma University Institute of Health Ethical Review Board with ref number (JUIH/IRB/309/23). Adults 18 years or older provided their own written consent when feasible, and those with impaired consciousness were enrolled with written consent from next of kin.
**Table 1  Demographic and clinical characteristics of ICU admitted adult patients at JUMC.**

| Variables | No sepsis (n = 137) | | | Sepsis (n = 148) | | | p-value sepsis vs. No sepsis |
|---|---|---|---|---|---|---|---|
| | Survivors (n = 119) | Non-survivors (n = 18) | p-value | Survivors (n = 94) | Non-survivors (n = 54) | p-value | |
| **Age, years, median (IQR)** | 35 (25–46) | 40.5 (28–64) | 0.10 | 30 (25–42) | 35 (25–50) | 0.42 | 0.37 |
| **Male, n (%)** | 79 (66.4) | 12 (66.7) | 0.60 | 48 (51.1) | 34 (63.0) | 0.10 | 0.06 |
| **ICU, n (%)** | | | | | | | |
| Medical | 27 (22.7) | 7 (38.9) | 0.1 | 23 (24.5) | 10 (18.5) | 0.69 | 0.81 |
| Surgical | 42 (35.3) | 8 (44.4) | | 36 (38.3) | 23 (42.6) | | |
| Emergency | 50 (42.0) | 3 (16.7) | | 35 (37.2) | 21 (38.9) | | |
| **Admission diagnosis, n (%)** | | | | | | | |
| Sepsis | 0 (0) | 0 (0) | | 94 (100) | 54 (100) | | |
| Respiratory disease | 6 (5) | 0 (0) | 0.42 | 41 (43.6) | 19 (35.2) | 0.31 | **<0.001** |
| Trauma | 11 (9.2) | 2 (11.1) | 0.53 | 10 (10.6) | 3 (5.6) | 0.23 | 0.83 |
| Heart Failure | 23 (19.3) | 5 (27.8) | 0.40 | 3 (3.2) | 5 (9.3) | 0.11 | **<0.001** |
| GI infection | 7 (5.9) | 2 (11.1) | 0.33 | 3 (3.2) | 6 (11.1) | **0.05** | 0.86 |
| Post-op monitoring | 20 (16.8) | 4 (22.2) | 0.38 | 4 (4.3) | 5 (9.3) | 0.19 | **0.003** |
| GBS | 9 (7.6) | 3 (16.7) | 0.19 | 3 (3.2) | 0 (0) | 0.25 | **0.01** |
| Others | 28 (23.5) | 4 (22.2) | 0.46 | 8 (8.5) | 4 (7.4) | 0.52 | **0.01** |
| **Site of infection, n (%)** | | | | | | | |
| Lung | 15 (12.6) | 3 (16.7) | 0.43 | 44 (46.8) | 28 (51.9) | 0.33 | **<0.001** |
| Abdomen | 15 (12.6) | 2 (11.1) | 0.60 | 16 (17.0) | 10 (18.5) | 0.49 | 0.22 |
| Urinary tract | 1 (0.8) | 1 (5.6) | 0.24 | 4 (4.3) | 1 (1.9) | 0.39 | 0.25 |
| Soft tissue | 6 (5) | 0 (0) | 0.42 | 7 (7.4) | 4 (7.4) | 0.63 | 0.20 |
| CNS | 10 (8.4) | 4 (22.2) | 0.09 | 8 (8.5) | 5 (9.3) | 0.54 | 0.41 |
| Other | 16 (13.4) | 5 (27.8) | 0.11 | 11 (11.7) | 3 (5.6) | 0.17 | 0.09 |
| **Comorbidities, n (%)** | | | | | | | |
| Respiratory disease | 13 (10.9) | 1 (5.6) | 0.42 | 34 (36.2) | 25 (46.3) | 0.15 | **<0.001** |
| Heart disease | 15 (12.6) | 3 (16.7) | 0.43 | 7 (7.4) | 4 (7.4) | 0.63 | 0.08 |
| Malaria | 5 (4.2) | 0 (0) | 0.48 | 23 (24.5) | 6 (11.1) | **0.05** | **<0.001** |
| Hypertension | 12 (10.1) | 4 (22.2) | 0.13 | 5 (5.3) | 3 (5.6) | 0.61 | **0.04** |
| Diabetes mellitus | 5 (4.2) | 6 (33.3) | **0.001** | 9 (9.6) | 5 (9.3) | 0.59 | 0.41 |
| Cancer | 8 (6.7) | 2 (11.1) | 0.38 | 3 (3.2) | 2 (3.7) | 0.60 | 0.11 |
| Other | 7 (5.9) | 0 (0) | 0.36 | 2 (2.1) | 3 (5.6) | 0.25 | 0.33 |
| **Organ failure, n (%)** | | | | | | | |
| Liver | 2 (1.7) | 1 (5.6) | 0.34 | 1 (1.1) | 4 (7.4) | 0.06 | 0.40 |
| Kidney | 5 (4.2) | 3 (16.7) | 0.07 | 8 (8.5) | 7 (13.0) | 0.27 | 0.13 |
| Lung | 15 (12.6) | 5 (27.8) | 0.09 | 33 (35.1) | 22 (40.7) | 0.30 | **<0.001** |
| Heart | 9 (7.6) | 2 (11.1) | 0.43 | 9 (9.6) | 7 (13.0) | 0.35 | 0.27 |
| Brain | 13 (10.9) | 3 (16.7) | 0.35 | 10 (10.6) | 6 (11.1) | 0.56 | 0.48 |
| Other | 10 (8.4) | 0 (0) | 0.23 | 8 (8.5) | 3 (5.6) | 0.37 | 0.57 |

**Table 1** (*continued*)

| Variables | No sepsis (*n* = 137) | | | Sepsis (*n* = 148) | | | *p*-value sepsis *vs.* No sepsis |
|---|---|---|---|---|---|---|---|
| | Survivors (*n* = 119) | Non-survivors (*n* = 18) | *p*-value | Survivors (*n* = 94) | Non-survivors (*n* = 54) | *p*-value | |
| **Number of organ failure, n (%)** | | | | | | | |
| None | 70 (58.8) | 8 (44.4) | 0.03 | 35 (37.2) | 16 (29.6) | 0.43 | **0.001** |
| One | 44 (37.0) | 7 (38.9) | | 50 (53.2) | 28 (51.9) | | |
| Two | 5 (4.2) | 2 (11.1) | | 8 (8.5) | 9 (16.7) | | |
| Three | 0 (0) | 1 (5.6) | | 1 (1.1) | 1 (1.9) | | |
| **Admission vital signs** | | | | | | | |
| Pulse rate | 100 (86–115) | 105 (90–120) | 0.24 | 110 (98–122) | 112.5 (104–137) | **0.02** | **<0.001** |
| Respiratory rate | 22 (20–26) | 24 (19–28) | 0.59 | 26 (20–30) | 26 (22–30) | 0.61 | **0.002** |
| **Interventions, n (%)** | | | | | | | |
| MV | 43 (36.1) | 10 (55.6) | 0.12 | 62 (65.9) | 41 (75.9) | 0.20 | **<0.001** |
| Vasopressor | 15 (12.6) | 1 (5.6) | 0.38 | 20 (21.3) | 16 (29.6) | 0.25 | **0.006** |
| Food supplement | 37 (31.1) | 6 (33.3) | 0.85 | 36 (38.3) | 21 (38.9) | 0.94 | 0.21 |
| Antibiotics | 64 (53.8) | 9 (50) | 0.76 | 61 (64.9) | 33 (61.1) | 0.75 | 0.07 |
| Other | 13 (10.9) | 2 (11.1) | 0.98 | 4 (4.3) | 2 (3.7) | 0.62 | **0.03** |
| **GCS** | 15 (10–15) | 15 (11–15) | 0.97 | 13.5 (7–15) | 7 (4–10) | **<0.001** | **<0.001** |
| **Oxygen saturation** | 96 (95–97) | 96 (94–97) | 0.65 | 96 (94–98) | 96 (92–97) | 0.40 | 0.33 |
| **Illness days before ICU, median (IQR)** | 2 (1–5) | 3 (2–10) | 0.06 | 2.5 (1–5) | 2 (1–7) | 0.85 | 0.88 |
| **In ICU days, median (IQR)** | 3 (2–5) | 3 (1–5) | 0.31 | 4 (3–7) | 2.5 (1–4) | **<0.001** | 0.57 |

**Notes.**

CNS, central nervous system; GBS, Guillain–Barré syndrome; GCS, Glascow coma scale; GI, gastrointestinal; HTN, hypertension; DM, diabetes mellitus; IQR, interquartile range; ICU, intensive care unit; MV, mechanical ventilator.

Statistically significant *p*-values (≤0.05) were bold in the tables.

# RESULTS

## Baseline characteristics of sepsis patients

During the one-year study period, 285 adults were admitted to the ICUs, of which 148 (51.9%) were diagnosed with sepsis at admission. Among the sepsis patients enrolled, 113 remained hospitalized until the third day, while 54 patients continued to the fifth day. Over three-fourths of admissions occurred in surgical and emergency ICUs. The in-ICU mortality rate for sepsis patients was 54 (36.5%) compared to 18 (13.1%) in adults without sepsis. Respiratory disease was the most common admission diagnosis among sepsis patients while heart failure was the leading diagnosis in non-sepsis patients. Respiratory tract infection and lung failure were the most frequently observed sites of infection and organ failure among ICU patients. More than half of the patients required mechanical ventilation and received antibiotic treatment. The median ICU stay of sepsis patients was three days (Table 1).

## Clinical scores at different days of admission

Five clinical scores—SIRS, qSOFA, UVA, MEWS, ad NEWS—were calculated for all ICU admitted patients across multiple days of admission. All scores were significantly different
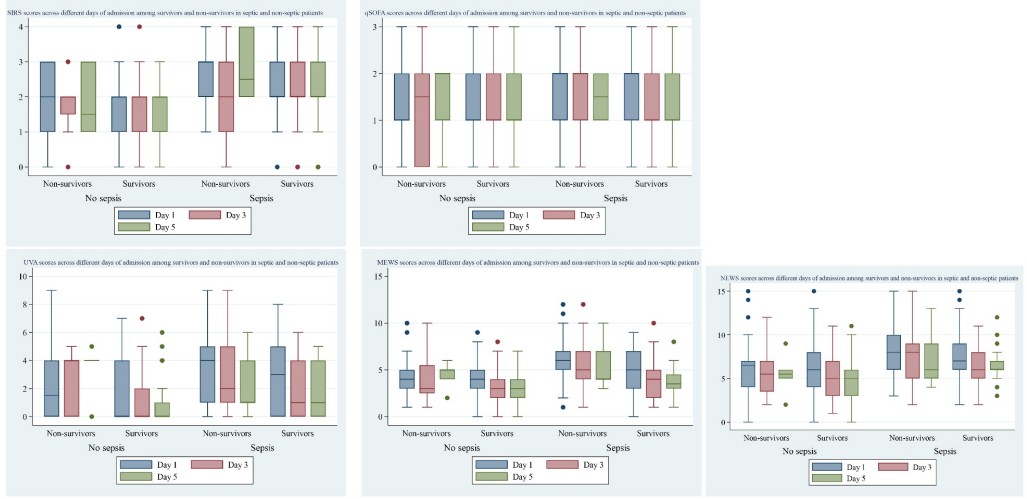

**Figure 1** Clinical scores across different days of admission by survival status in patients with sepsis and without sepsis.

between patients with sepsis and without sepsis. Among sepsis patients, all scores except SIRS were significantly different between survivors and non-survivors by the third day of admission (Fig. 1).

## Performance of clinical scores to predict sepsis

The performance of the five clinical scores to discriminate between patients with and without sepsis was not significantly different at the day of admission ($p = 0.16$). However, on days 3 and 5, the scores demonstrated statistically significant difference. On the day of admission, the MEWS score performed well, compared to the others, with an AUC of 0.67 (95% CI [0.61–0.73]). By day 3 the NEWS score performed well (AUC = 0.67, 95% CI [0.59–0.74]), while the SIRS score showed better performance on day 5 (AUC = 0.69, 95% CI [0.59–0.79]). Among the scores, the qSOFA score exhibited the lowest discriminatory power (Fig. 2).

With a cut-off of ≥2, the SIRS score demonstrated high sensitivity (83.1%) but low specificity (33.6%). The qSOFA score, at a cut-off of ≥2, showed moderate sensitivity (58.1%) and specificity (59.1%). The UVA score, with a cut-off of ≥5, had lower sensitivity (33.1%) but high specificity (86.8%). The MEWS score at a cut-off of ≥5 provided a performance with 64.2% sensitivity and 62.5% specificity. Meanwhile, the NEWS score, also with a cut-off of ≥5, exhibited the highest sensitivity (89.8%) but lower specificity (36.8%) (Table 2).

## Performance of clinical scores to predict mortality in sepsis patients

The predictive performance of the five clinical scores for mortality in sepsis patients showed significant differences by the third day of admission ($p = 0.01$). Among them, the MEWS score consistently performed well across all days, with an AUC of 0.62 (95% CI [0.53–0.72])
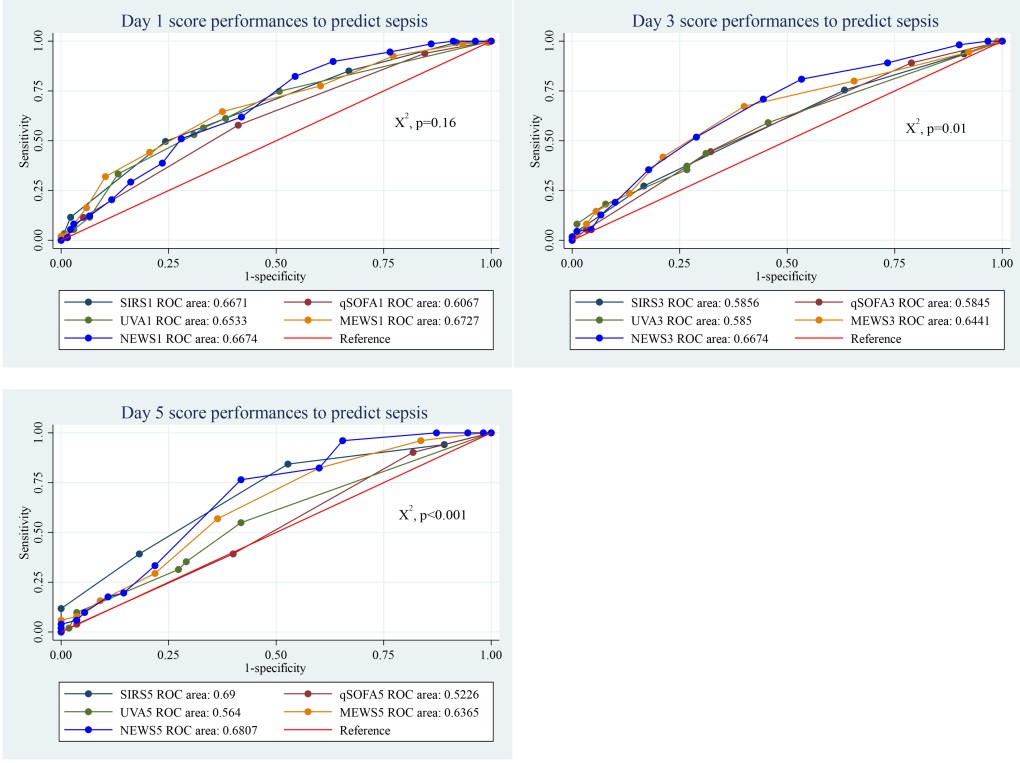

**Figure 2  Performance of clinical scores to predict sepsis at different days of admission.**

on day 1, 0.71 (95% CI [0.60–0.81]) on day 3, and 0.75 (95% CI [0.57–0.93]) on day 5 (Fig. 3).

At day 1, the sensitivity and specificity of SIRS to predict mortality with a cut-off value of ≥2 was 88.9% and 17.0% respectively. Similarly, the sensitivity and specificity of qSOFA to predict mortality with a cut-off point of ≥2 were 68.5% and 47.9%, respectively. The UVA score at day 1 with ≥5 cut-off point, predicted mortality with 38.9% sensitivity and 70.2% specificity. The MEWS sensitivity and specificity at day 1 with ≥5 cut-off point was 77.8% and 43.6%, respectively. It was 92.6% and 11.8% for NEWS with ≥5 cut-off point (Table 3).

## Performance of clinical scores to predict culture positivity in sepsis patients

By day 3, clinical scores showed a statistically significant difference in predicting culture-positive sepsis patients ($p = 0.03$). The MEWS score performed well across all observed days, compared to the other scores. Its AUC value was 0.65 (95% CI [0.50–0.80]), on day 1, 0.69 (96% CI [0.55–0.84]), on day 3, and 0.58 (95% CI [0.37–0.79]) on day 5 (Fig. 4). With a cut-off of ≥2, the SIRS score exhibited high sensitivity (86.7%) but low specificity (20.8%). The qSOFA score, at a cut-off of ≥2, showed moderate sensitivity (60.0%) and specificity (57.3%). The UVA score, with a cut-off of ≥5, had lower sensitivity (40.0%) but high specificity (86.3%). The MEWS score at a cut-off of ≥5 demonstrated high sensitivity

**Table 2** Sensitivity and specificity of clinical scores to identify sepsis with cut-offs on each days of admission.

| Days | Cut-off | Sensitivity | Specificity | Correctly classified | LR+ | LR- |
|------|---------|-------------|-------------|----------------------|-----|-----|
| **SIRS** | | | | | | |
| 1 | ≥2 | 85.14% | 33.58% | 60.35% | 1.2817 | 0.4427 |
| 3 | ≥2 | 73.45% | 38.04% | 57.56% | 1.1855 | 0.6979 |
| 5 | ≥2 | 81.48% | 47.37% | 63.96% | 1.5481 | 0.3909 |
| **qSOFA** | | | | | | |
| 1 | (≥2) | 58.11% | 59.12% | 58.60% | 1.4216 | 0.7085 |
| 3 | (≥2) | 55.17% | 44.55% | 67.74% | 1.3809 | 0.8186 |
| 5 | (≥2) | 37.74% | 58.93% | 48.62% | 0.9188 | 1.0566 |
| **UVA** | | | | | | |
| 1 | (≥5) | 33.11% | 86.76% | 58.80% | 2.5015 | 0.7710 |
| 3 | (≥5) | 18.18% | 92.22% | 51.50% | 2.3377 | 0.8872 |
| 5 | (≥5) | 9.26% | 94.83% | 53.57% | 1.7901 | 0.9569 |
| **MEWS** | | | | | | |
| 1 | (≥5) | 64.19% | 62.50% | 63.38% | 1.7117 | 0.5730 |
| 3 | (≥5) | 41.82% | 78.89% | 58.50% | 1.9809 | 0.7375 |
| 5 | (≥5) | 28.30% | 76.79% | 53.21% | 1.2192 | 0.9337 |
| **NEWS** | | | | | | |
| 1 | (≥5) | 89.80% | 36.76% | 64.31% | 1.4200 | 0.2776 |
| 3 | (≥5) | 80.91% | 46.67% | 65.50% | 1.5170 | 0.4091 |
| 5 | (≥5) | 81.13% | 41.07% | 60.55% | 1.3768 | 0.4594 |

Notes.
LR, likelihood ratio; MEWS, modified early warning score; NEWS, national early warning score; SIRS, systemic inflammatory response syndrome; qSOFA, quick sequential organ failure assessment; UVA, universal vital assessment.

(80.0%) and moderate specificity (49.5%). Meanwhile, the NEWS score, also with a cut-off of ≥5, exhibited high sensitivity (86.7%) but low specificity (24.2%) (Table 4).

## Survival analysis

Based on the Kaplan–Meier survival analysis presented in Fig. 5, the median survival time for sepsis patients in the ICU was 13 days. The probability of sepsis patients surviving 14 days or longer in the ICU was 40%, compared to 69% for patients without sepsis. The log-rank test showed a statistically significant difference in survival between patients with and without sepsis ($p < 0.001$). Additionally, the Cox proportional hazard model (Table 5) showed that every added unit of GCS reduced the hazard of death by 10% (HR = 0.90; 95% CI [0.84–0.97]; $p = 0.005$).

## DISCUSSION

Sepsis significantly contributes to ICU admission and mortality. This study presented that 148 (51.9%) of ICU admitted adults were diagnosed with sepsis, of which 54 (36.5%) patients died in the ICU. The probability of sepsis patients surviving longer days in ICU was significantly lower compared to patients without sepsis. Early identification of sepsis and related outcomes using accessible clinical and laboratory tools is vital to mitigate its effect on ICU admitted patients. The present study evaluated the performance of five

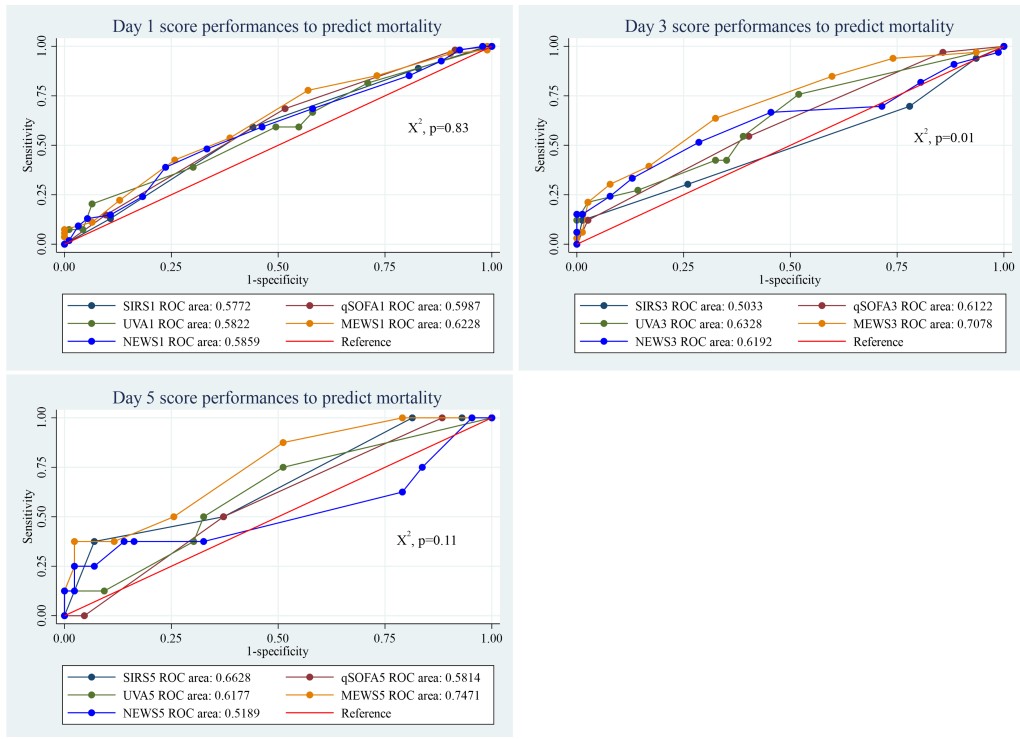

**Figure 3** Performance of clinical scores to predict mortality in sepsis patients at different days of admission.

clinical scores to discriminate between patients with and without sepsis, as well as to predict mortality and culture positivity in ICU admitted septic adults. On each day of admission, the MEWS score exhibited the best performance, compared to other scores, in predicting mortality and culture positivity in sepsis patients. However, at the third and fifth day of admission, the NEWS and SIRS performed better than MEWS in discriminating between patient with and without sepsis.

The current study revealed that the most common admission diagnosis and comorbidity in sepsis patients was respiratory disease. Sepsis and respiratory disease are closely linked, leading to a significant increase in mortality. For instance, sepsis induced acute respiratory distress syndrome (ARDS) causes wide spread alveolar damage and organ failure (*Gong et al., 2022*), which results in high mortality in critically ill patients (*Kim & Hong, 2016*). Our study also revealed that lung failure was most common than other organ failures in sepsis patients. Moreover, malaria infection was found to be higher (19.6%) among sepsis patients admitted to the ICUs. The mortality of patients with malaria and concomitant sepsis was reported to be higher, particularly in resource limited settings (*Njim et al., 2018*).

In this study, the SIRS score exhibited irregular pattern of performance over the days. The non-specific nature of the SIRS criteria could explain this pattern. Since SIRS can be triggered by various conditions other than sepsis, the score might increase in patients with different types of inflammation or other medical conditions, leading to its less specific

**Table 3** Sensitivity and specificity of clinical scores to predict mortality in sepsis with cut-offs on each day of admission.

| Days | Cut-off | Sensitivity | Specificity | Correctly classified | LR+ | LR- |
|---|---|---|---|---|---|---|
| **SIRS** | | | | | | |
| 1 | (≥2) | 88.89% | 17.02% | 43.24% | 1.0712 | 0.6528 |
| 3 | (≥2) | 67.65% | 24.05% | 37.17% | 0.8907 | 1.3452 |
| 5 | (≥2) | 100.00% | 21.74% | 33.33% | 1.2778 | 0.0000 |
| **qSOFA** | | | | | | |
| 1 | (≥2) | 68.52% | 47.87% | 55.41% | 1.3144 | 0.6576 |
| 3 | (≥2) | 54.55% | 59.74% | 58.18% | 1.3548 | 0.7609 |
| 5 | (≥2) | 50.00% | 64.44% | 62.26% | 1.4062 | 0.7759 |
| **UVA** | | | | | | |
| 1 | (≥5) | 38.89% | 70.21% | 58.78% | 1.3056 | 0.8704 |
| 3 | (≥5) | 27.27% | 85.71% | 68.18% | 1.9091 | 0.8485 |
| 5 | (≥5) | 11.11% | 91.11% | 77.78% | 1.2500 | 0.9756 |
| **MEWS** | | | | | | |
| 1 | (≥5) | 77.78% | 43.62% | 56.08% | 1.3795 | 0.5095 |
| 3 | (≥5) | 63.64% | 67.53% | 66.36% | 1.9600 | 0.5385 |
| 5 | (≥5) | 44.44% | 75.00% | 69.81% | 1.7778 | 0.7407 |
| **NEWS** | | | | | | |
| 1 | (≥5) | 92.59% | 11.83% | 41.50% | 1.0501 | 0.6263 |
| 3 | (≥5) | 81.82% | 19.48% | 38.18% | 1.0161 | 0.9333 |
| 5 | (≥5) | 77.78% | 18.18% | 28.30% | 0.9506 | 1.2222 |

**Notes.**

LR, likelihood ratio; MEWS, modified early warning score; NEWS, national early warning score; SIRS, systemic inflammatory response syndrome; qSOFA, quick sequential organ failure assessment; UVA, universal vital assessment.

performance in predicting sepsis (*Vincent et al., 2013*). The MEWS demonstrated better performance in identifying sepsis on the day of admission with an AUC of 0.672, but its ability declined over subsequent days. The qSOFA and UVA score had lower performance on the day of admission, and their performance continued to decline over the following days. A decline in performance over the subsequent days might suggest that the sepsis condition is improving over time, causing the scores to normalize. It is worth noting that the sepsis, the outcome variable in this case, was diagnosed at the day of admission. However, the NEWS score exhibited increasing performance in the following days after admission with an AUC of 0.681 at day 5 of admission. This could be due to the additional parameters included in the NEWS score, such as oxygen saturation and supplemental oxygen, which may have contributed to its differing performance. The NEWS score also showed better sensitivity at all days with cut-off of five or greater. This is consistent with a previous study that evaluated the performance of NEWS score to predict sepsis and reported an AUC of 0.685 (*Thodphetch et al., 2021*). Most of the previous studies were more focused on patients at general wards and emergency department to evaluate the performance of early warning scores in predicting ICU transfer and mortality (*Thodphetch et al., 2021*; *Sabir et al., 2020*; *Usul et al., 2021*; *Lee & Choi, 2014*).

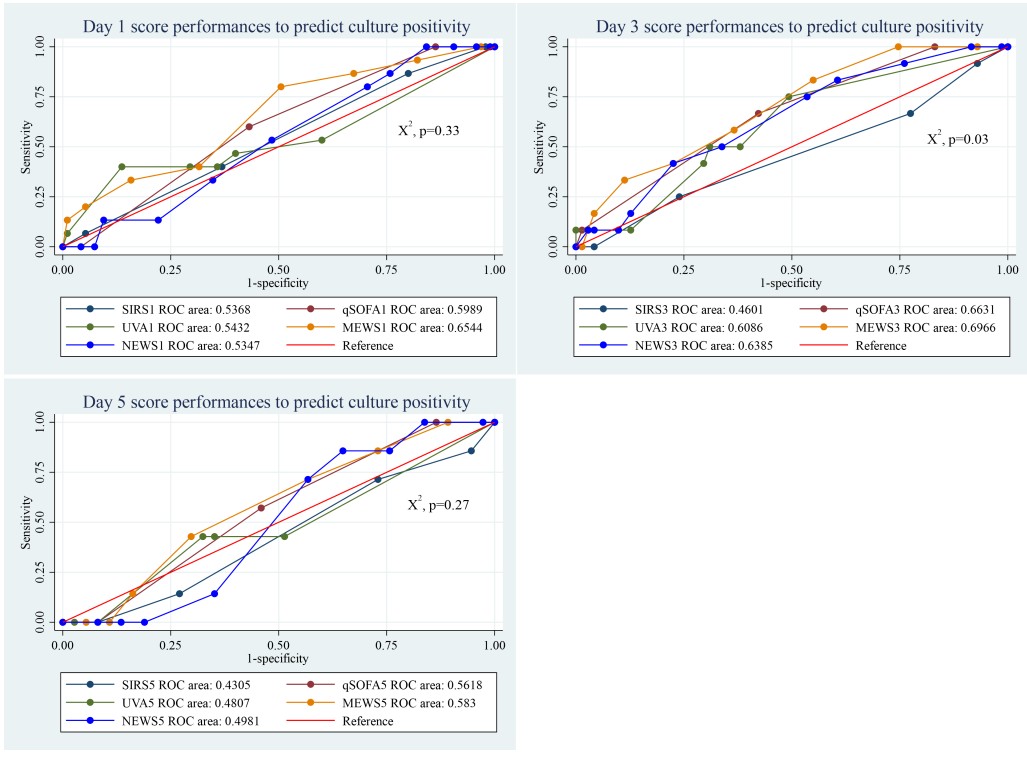

**Figure 4** **Performance of clinical scores to predict culture positivity in sepsis patients at different days of admission.**

For predicting in-ICU mortality, the MEWS score showed the highest performance among the clinical scores evaluated, with its efficiency increasing over the subsequent days, reaching an AUC of 0.747 on day five of admission. Previous study in Thailand that assessed the performance of different clinical scores for predicting in-ICU mortality reported a higher AUC (0.861) for the MEWS score (*Khwannimit, Bhurayanontachai & Vattanavanit, 2019*). Similarly, a multi-center study in Iran involving general ICU admitted patients showed that the AUC of MEWS was 0.881 (*Mahmoodpoor et al., 2022*). The performance of the other scores in this study was lower and inconsistent over the days. In contrary to our study, the study in Iran reported a higher AUC (0.916) for NEWS in predicting in-ICU mortality. In fact, at the time of admission, the NEWS score demonstrated the highest sensitivity (92.6%) for predicting in-ICU mortality when using a cut-off of ≥5, in our study. However, its specificity was quite low at 11.8%. Differences in healthcare infrastructure, patient demographics, and cut-off values used might have caused the difference in the predictive performance of the scores across different studies.

Regarding the prediction of culture positivity in sepsis patients, the MEWS consistently demonstrated better performance over the days. However, the AUC of MEWS was better on day three of admission. This pattern might suggest that patients with positive cultures may initially have higher MEWS scores, which then decrease over time as the infection is treated and their condition improves. The sensitivity and specificity of MEWS with cut-off

**Table 4** Sensitivity and specificity of clinical scores to predict culture positivity in sepsis with cut-offs on each day of admission.

| Days | Cut-off | Sensitivity | Specificity | Correctly classified | LR+ | LR- |
|------|---------|-------------|-------------|----------------------|------|------|
| **SIRS** | | | | | | |
| 1 | (≥2) | 86.67% | 20.83% | 29.73% | 1.0947 | 0.6400 |
| 3 | (≥2) | 66.67% | 24.66% | 30.59% | 0.8848 | 1.3519 |
| 5 | (≥2) | 71.43% | 25.64% | 32.61% | 0.9606 | 1.1143 |
| **qSOFA** | | | | | | |
| 1 | (≥2) | 60.00% | 57.29% | 57.66% | 1.4049 | 0.6982 |
| 3 | (≥2) | 66.67% | 56.94% | 58.33% | 1.5484 | 0.5854 |
| 5 | (≥2) | 57.14% | 52.63% | 53.33% | 1.2063 | 0.8143 |
| **UVA** | | | | | | |
| 1 | (≥5) | 40.00% | 86.32% | 80.00% | 2.9231 | 0.6951 |
| 3 | (≥5) | 8.33% | 87.32% | 75.90% | 0.6574 | 1.0497 |
| 5 | (≥5) | 0.00% | 92.68% | 79.17% | 0.0000 | 1.0789 |
| **MEWS** | | | | | | |
| 1 | (≥5) | 80.00% | 49.47% | 53.64% | 1.5833 | 0.4043 |
| 3 | (≥5) | 58.33% | 63.38% | 62.65% | 1.5929 | 0.6574 |
| 5 | (≥5) | 42.86% | 70.00% | 65.96% | 1.4286 | 0.8163 |
| **NEWS** | | | | | | |
| 1 | (≥5) | 86.67% | 24.21% | 32.73% | 1.1435 | 0.5507 |
| 3 | (≥5) | 83.33% | 39.44% | 45.78% | 1.3760 | 0.4226 |
| 5 | (≥5) | 85.71% | 35.90% | 43.48% | 1.3371 | 0.3980 |

Notes.

LR, likelihood ratio; MEWS, modified early warning score; NEWS, national early warning score; SIRS, systemic inflammatory response syndrome; qSOFA, quick sequential organ failure assessment; UVA, universal vital assessment.

**Table 5** Cox proportional hazard regression analysis in sepsis patients.

| Variables | Hazard ratio | $P$-value | 95% CI |
|-----------|--------------|-----------|--------|
| Age | 1.01 | 0.087 | 1.00–1.03 |
| Sex, F | 0.81 | 0.470 | 0.46–1.43 |
| Organ failure | 1.00 | 0.974 | 0.53–1.86 |
| Comorbidities | 1.15 | 0.688 | 0.59–2.22 |
| Mechanical ventilation use | 0.56 | 0.159 | 0.25–1.25 |
| Vasopressor use | 1.42 | 0.255 | 0.78–2.58 |
| Glasgow coma scale | 0.90 | **0.005** | 0.84–0.97 |

Notes.

Statistically significant $p$-values (≤0.05) were bold in the tables.

of ≥5 was 80% and 49.5%, respectively. This trend underscores the valuable role MEWS can play in aiding early decision-making and monitoring the progression and recovery of sepsis.

Another important finding in this study was the survival analysis which reported the median in-ICU survival of 13 days in sepsis patients. This aligns with a previous study that reported the median survival of 11 days for low-risk and 15 days for high-risk groups (*Li et al., 2020*). According to the log-rank test, the survival between patients with and without

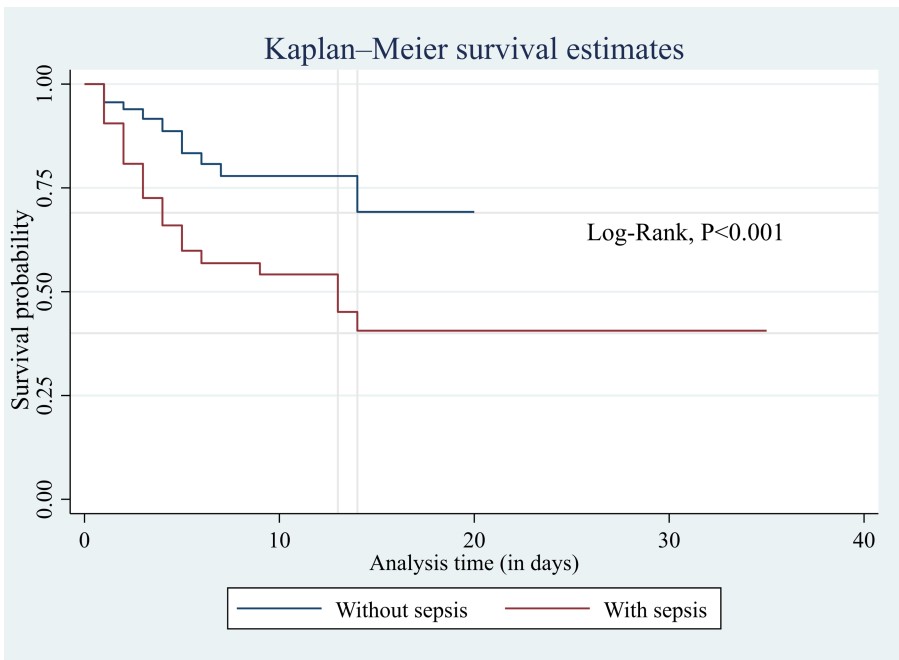

**Figure 5** **Kaplan–Meier survival analysis to show survival probability between sepsis and non-sepsis.**

sepsis was statistically significantly different. This is in agreement with a finding that reported a higher in-ICU mortality in sepsis patients compared to patients without sepsis (*Hodgson et al., 2022*). Patients without sepsis had a 75% higher probability of surviving 14 days or more in the ICU compared to those with sepsis. Our study reported that a unit increase in GCS reduced the hazard of death by 10% in sepsis patients. This finding aligns with previous study that highlight the prognostic significance of the level of consciousness at admission, as measured by the GCS, in predicting ICU and hospital mortality (*Bastos et al., 1993*).

The strength of this study is that it included patients from various ICUs, such as medical, surgical, and emergency ICUs. It also provided clinical scores for subsequent days after admission, in addition to the scores at admission. This approach enabled tracking the trend of the scores over the following days. However, the study has several limitations. Firstly, the sepsis diagnosis did not utilize the sepsis-3 definition as currently proposed. Additionally, as a single-center study in an ICUs, its findings may not be generalizable to the wider population. Patients longer stay in the ICUs might also skew the survival estimate. Thus, the findings should be validated by future multi-center studies that took these limitations into account.

## CONCLUSIONS

The MEWS score demonstrated good discriminatory power in identifying sepsis patients at ICU admission and those with culture-positive results. It also proved to be a better predictor of ICU mortality compared to other scores. The simplicity of the MEWS score,

in contrast to other scores like SOFA, APACHE, and SAPS, makes it advantageous for routine use in resource-limited settings. Additionally, the study found that sepsis patients had higher mortality rates, which decreased as their Glasgow Coma Scale (GCS) scores increased. Prioritizing efforts to mitigate the mortality of sepsis patients admitted to ICUs is essential. This can include timely recognition of sepsis, prompt administration of appropriate antibiotics, close monitoring of vital signs and organ function, and aggressive supportive care. Integration of MEWS into routine ICU protocols of resource-limited areas and training healthcare workers for its effective use can significantly improve patient outcomes by enabling early identification and risk-stratification of critically ill patients.

### Funding
The study was supported by the institute of health research and innovation director office, Jimma University with Mega Research Fund Scheme (2022-2025). The funders had no role in study design, data collection and analysis, decision to publish, or preparation of the manuscript.

### Grant Disclosures
The following grant information was disclosed by the authors:
The institute of health research and innovation director office, Jimma University with Mega Research Fund Scheme (2022-2025).

### Competing Interests
The authors declare there are no competing interests.

### Author Contributions
- Girum Tesfaye Kiya conceived and designed the experiments, performed the experiments, analyzed the data, prepared figures and/or tables, authored or reviewed drafts of the article, and approved the final draft.
- Zeleke Mekonnen performed the experiments, authored or reviewed drafts of the article, and approved the final draft.
- Elsah Tegene Asefa performed the experiments, authored or reviewed drafts of the article, and approved the final draft.
- Edosa Kejela performed the experiments, authored or reviewed drafts of the article, and approved the final draft.
- Edosa Tadasa performed the experiments, authored or reviewed drafts of the article, and approved the final draft.
- Esayas Kebede Gudina performed the experiments, authored or reviewed drafts of the article, and approved the final draft.
- Tilahun Yemane performed the experiments, authored or reviewed drafts of the article, and approved the final draft.
- Gemeda Abebe performed the experiments, authored or reviewed drafts of the article, and approved the final draft.

## Human Ethics

The following information was supplied relating to ethical approvals (*i.e.*, approving body and any reference numbers):

This study was approved by the Jimma University Institute of Health Ethical Review Board with ref number (JUIH/IRB/309/23).

## Data Availability

The raw measurements are available in the Supplementary File.

## Supplemental Information

Supplemental information for this article can be found online at http://dx.doi.org/10.7717/peerj.20109#supplemental-information.

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
