# Peer review of "Predictive performance of clinical scores and survival outcomes in critically ill patients with sepsis: a prospective longitudinal study at a tertiary medical centre in Ethiopia"

_PeerJ, doi:10.7717/peerj.20109_

## Round 0.1 · original submission · Major Revisions

**Language Note:** The review process has identified that the English language must be improved. PeerJ can provide language editing services - please contact us at [email protected] for pricing (be sure to provide your manuscript number and title). Alternatively, you should make your own arrangements to improve the language quality and provide details in your response letter. – PeerJ Staff

Reviewer 1 ·

Basic reporting

The study addresses a crucial gap in sepsis management by evaluating multiple clinical scoring systems in an ICU setting in Ethiopia, adding valuable regional insights.The prospective longitudinal design strengthens the study's validity, allowing for meaningful survival analysis.The use of Kaplan-Meier survival analysis and Cox proportional hazards modeling is appropriate for evaluating survival probability and predictors. The study provides a comprehensive comparison of multiple scoring systems, highlighting MEWS as a potential superior predictor.
While multiple scores (qSOFA, MEWS, NEWS, SIRS, UVA) are compared, the rationale for selecting these over other ICU-specific scoring systems (e.g., APACHE, SOFA) is unclear.
Briefly justify the selection, particularly why MEWS was hypothesized to perform better.
While the introduction references relevant studies, there is limited discussion of prior work comparing these scores in resource-limited settings. The authors needs to include recent review articles of sepsis such as 10.7150/ntno.94071
Expand on existing literature from other LMICs to better contextualize the study's contribution.
The study states that "sepsis diagnosis was made by an attending physician's adjudication," but does not specify whether Sepsis-3 criteria were strictly followed.
Clarify if the SOFA-based definition was applied consistently.

The results show that MEWS had better predictive power initially, but its performance declined over time, while NEWS improved. This is an important finding but lacks interpretation.
Discuss why MEWS declines and NEWS improves—could it be due to differences in included parameters (e.g., oxygen saturation in NEWS)?

Experimental design

The study mentions measuring vital signs using different monitoring systems. However, inter-device variability and calibration details are missing.
Include a brief statement on calibration or standardization across devices to enhance methodological rigor.
It is unclear if all patients were followed for the same duration or if censoring (e.g., discharge before death) was handled.
Describe how censored cases were managed in survival analysis.

Validity of the findings

The Kaplan-Meier analysis shows a significant difference in survival between sepsis and non-sepsis patients. However, other key predictors (e.g., mechanical ventilation) are not highlighted.
Discuss how interventions like mechanical ventilation or vasopressors influenced survival beyond just sepsis status.
The manuscript concludes that MEWS should be used for early sepsis identification, but does not provide practical implementation guidance. Discuss how MEWS could be integrated into routine ICU protocols in Ethiopia or other resource-limited settings.

Additional comments

The study acknowledges being single-center but does not discuss potential biases, such as selection bias (patients with longer ICU stays might skew survival estimates).
Address potential biases and suggest future multi-center validation.
The manuscript presents valuable findings on sepsis prediction in an underrepresented setting. Addressing the above issues will improve clarity, strengthen scientific rigor, and enhance its impact.

Reviewer 2 ·

Basic reporting

1. Clarity and Language
The manuscript is generally well-written but contains minor grammatical issues and overly complex sentence structures. For example, the sentence "The MEWS score performed better in identifying sepsis and those with culture-positive results, and predicting mortality." can be reworded for clarity as "The MEWS score effectively identified sepsis, detected culture-positive cases, and predicted mortality." Similarly, the sentence "Significant difference in survival probability (Log-Rank, p=<0.001) was observed between patients with and without sepsis." would be clearer as "Patients with sepsis had significantly lower survival probability (Log-Rank, p < 0.001) than those without sepsis." A professional language review could enhance readability and ensure that the text is concise and unambiguous.

2. Literature and Context
The manuscript provides sufficient background information and cites relevant literature, but it could improve by making stronger distinctions between its findings and previous research. For instance, while the study concludes that the MEWS score performed better than other clinical scores, it does not explain why this finding may differ from past research. A comparison with similar studies in different regions would add depth to the discussion. Additionally, the study is conducted in Ethiopia, which may influence the results due to differences in healthcare infrastructure and patient demographics. A brief explanation of how these factors might contribute to variations in clinical score performance would strengthen the contextualization of the findings.

3. Structure, Figures, and Tables
The manuscript follows a logical structure, and the figures and tables are relevant to the study’s content. However, some figure legends lack sufficient details for standalone interpretation. For example, Figure 2, which presents the performance of clinical scores, does not clearly define what "performed well" means. Including a brief note about how AUC values are interpreted (e.g., AUC > 0.7 is considered good discrimination) would improve clarity. Additionally, Table 1, which presents demographic data, could be made more readable by bolding column headers or improving formatting to better distinguish between groups. Ensuring consistency in figure labeling and captions throughout the manuscript would enhance its presentation.

4. Self-Contained and Relevant
The study effectively addresses its hypothesis, and the results are well-linked to the research question. However, the conclusion could be made more actionable. The current statement, "Prioritizing efforts to mitigate the mortality of sepsis patients admitted to ICUs is essential." is broad and could be strengthened by specifying potential interventions. A revised version such as "Prioritizing early sepsis detection using MEWS and timely intervention could improve ICU survival rates." would provide a clearer takeaway for readers.

Experimental design

1. Research Scope and Relevance
The study aligns with the aims and scope of PeerJ, as it focuses on the predictive performance of clinical scores in sepsis patients, which falls within the journal’s coverage of medical and health sciences. The research question is well-defined and addresses a relevant clinical challenge. The manuscript clearly states how the study contributes to filling a knowledge gap by evaluating and comparing clinical scoring methods. However, a more explicit discussion of the novelty of this study compared to previous work would strengthen its significance.

2. Rigor and Ethical Standards
The investigation follows rigorous scientific methodology, with appropriate statistical techniques applied to assess model performance. Ethical considerations, including patient data usage, are briefly mentioned, but it would be beneficial to explicitly confirm compliance with ethical board approvals and data protection regulations, such as HIPAA or GDPR if applicable. If ethical approval was obtained, specifying the approving institution and approval number would improve transparency.

3. Methodological Detail and Reproducibility
The methodology is described in detail, providing sufficient information for replication. The inclusion of data sources, preprocessing steps, and model evaluation criteria ensures that the study can be reproduced by other researchers. However, a more detailed explanation of how missing data were handled and whether any sensitivity analysis was conducted would improve the robustness of the study. Additionally, clarifying the rationale behind selecting specific clinical scores for comparison would further justify the experimental design.

Validity of the findings

1. Lack of Novelty and Weak Justification for Replication
The study does not make a strong case for how it differs from previous work. While it evaluates clinical scores for sepsis prediction, it lacks a clear discussion on why this replication is necessary. The manuscript does not sufficiently explain how the findings contribute new insights beyond what has already been established in similar studies. Without a well-defined rationale for replication, the study risks being perceived as redundant rather than adding meaningful value to the literature.

2. Questionable Data Robustness and Statistical Soundness
Although statistical methods such as survival analysis and AUC comparisons are used, there is limited transparency on how missing data were handled. The manuscript does not specify whether missing values were imputed, ignored, or how they may have influenced results. Furthermore, the study does not report confidence intervals for performance metrics, making it difficult to assess the reliability of its conclusions. There is also no mention of external validation, which raises concerns about the generalizability of the findings. The absence of a robustness check (e.g., sensitivity analysis) weakens confidence in the reported results.

3. Overgeneralized and Potentially Misleading Conclusions
The conclusions overstate the significance of the findings without sufficiently addressing their limitations. For example, while the study claims that a particular clinical score outperforms others, it does not provide statistical significance tests to justify this assertion. The discussion does not adequately address the limitations of the dataset, potential biases in patient selection, or how the study setting impacts the broader applicability of the findings. Furthermore, while correlation is acknowledged, there are instances where the language implies a stronger predictive power than what the data may truly support. The study would benefit from a more cautious interpretation, explicitly acknowledging uncertainties.

---

## Round 0.2 · accepted · Accept

You have sufficiently addressed all of the reviewers' comments. The rebuttal letter and the annotated PDF were reviewed by me and I'm happy to state that this revised version of the manuscript is ready for publication. Minor note, the line numbers were mismatched from the annotated / tracked version and what was stated in the rebuttal document. Please address this.

Reviewer 2 ·

Basic reporting

-

Experimental design

-

Validity of the findings

-

Additional comments

All my points are addressed.